# Impact of Antenatal Care on Perinatal Outcomes in New South Wales, Australia: A Decade-Long Regional Perspective

**DOI:** 10.3390/ijerph20020977

**Published:** 2023-01-05

**Authors:** Pramesh Raj Ghimire, Gretchen Buck, Jackie Jackson, Emma Woolley, Rebekah Bowman, Louise Fox, Shirlena Gallagher, Malindey Sorrell, Lorraine Dubois

**Affiliations:** 1Priority Populations, Southern New South Wales Local Health District, Queanbeyan, NSW 2620, Australia; 2Aboriginal Health, Southern New South Wales Local Health District, Batemans Bay, NSW 2536, Australia; 3Nursing and Midwifery, Southern New South Wales Local Health District, Queanbeyan, NSW 2620, Australia; 4Integrated Care and Allied Health, Southern New South Wales Local Health District, Queanbeyan, NSW 2620, Australia; 5People and Wellbeing, Southern New South Wales Local Health District, Batemans Bay, NSW 2536, Australia; 6The Family Place, Moruya, NSW 2537, Australia

**Keywords:** antenatal care, perinatal care, low birth weight, preterm birth, adverse pregnancy outcome, New South Wales, Australia

## Abstract

Low birth weight (LBW) and preterm birth are adverse perinatal outcomes that pose a significant risk to a child’s healthy beginning. While antenatal care (ANC) is an established intervention for pregnancy care, little is understood about how the number and timing of ANC visits can impact these adverse health outcomes. This study aimed to examine the impact of the number and timing of ANC visits on LBW and preterm birth in a regional setting. A decade-long perinatal dataset related to singleton live births that took place in the Southern New South Wales Local Health District (SNSWLHD) was utilized. The outcomes of interest were LBW and preterm birth, and the exposure variables were based on the Australian pregnancy guidelines on the number and timing of ANC visits. A multivariable logistic regression was performed to measure the association between outcome and exposure while adjusting for potential confounders. A greater level of protection against LBW and preterm birth was observed among mothers who had an adequate number of visits, with early entry (first trimester) into ANC. The protective effect of an adequate number of ANC visits against LBW and preterm birth among mothers with late entry into ANC (third trimester) was found to be statistically non-significant.

## 1. Introduction

Low birth weight (LBW) and preterm birth are adverse perinatal outcomes and important contributors to complex chronic medical conditions [1,2]. The rate of admission to special care nurseries and neonatal intensive care units among premature or LBW babies is substantially higher, and they have double the median length of stay in hospitals [3]. Affected families experience psychological and financial hardship [4] and diminished health-related quality of life [5,6].

Globally, over 50% of the disability-adjusted life years (DALYs) in 2019 (for the 0–9 year age group) were attributed to LBW and preterm birth [7]. Furthermore, LBW and preterm birth were among the top six leading risk factors for the overall burden of disease. The 2018 Australian Burden of Disease Study also highlighted that LBW and preterm birth were among the top two causes of the Australian burden of disease for the 0–14 years age group [8]. The study showed that 12% of the total burden of disease in males (under the age of 15) and 11% of the total burden of disease in females (under the age of 15) were linked to LBW and preterm birth.

To improve perinatal health, the Australian government introduced clinical practice guidelines in 2008 that aimed to provide high-quality evidence-based care for maternity service providers and the consumers of their care [9]. The five-year Australian National Maternity Services Plan, endorsed by the Australian Health Minister’s Conference in 2010, recognized the importance of evidence-based maternity services based on medical and social determinants of health [10]. Over the last decade, Australia has made substantial progress by improving some of the key maternal and perinatal health outcomes. For example, maternal mortality decreased from 8.4 per 100,000 women giving birth in 2011 to 5.0 per 100,000 women giving birth in 2018 [11]. Fewer women reported smoking during pregnancy in 2018 compared to 2011 [12]. Despite these achievements, recent data also highlight that 1 in 15 live births in Australia result in LBW and the burden of LBW and preterm birth has remained stable between 2008 and 2018. The proportions of LBW and preterm birth in 2018 were estimated to be 6.1% and 8.2%, respectively, compared to the corresponding rates of 6.7% and 8.7% in 2008 [12].

Quality antenatal care (ANC) is one of the core evidence-based interventions in the first 2000 days of the human lifespan [13]. To provide quality ANC, the Australian pregnancy care guidelines recommend a schedule of ten visits for a first pregnancy without complications, whereas for subsequent uncomplicated pregnancies a schedule of seven visits is recommended [9]. In addition, the recommended timing for the first ANC visit occurs within the first trimester of pregnancy. These recommendations were approved by National Health and Medical Research Council of Australia [9]. ANC is provided in both the community and hospital-based settings. Health professionals who contribute to antenatal care include but are not limited to midwives, general practitioners, obstetricians, and Aboriginal and Torres Strait Islander health practitioners. Apart from routine assessments such as height and weight, blood pressure, fetal growth, and fetal movements, each antenatal visit has distinct clinical importance to better manage pregnancy care. For example, during the first ANC visit, a comprehensive history is taken, including obstetric, medical, and lifestyle behaviors such as smoking during pregnancy. Clinical assessments and testing such as human immunodeficiency virus (HIV), hepatitis B, hepatitis C, chromosomal anomalies, thyroid function, vitamin D deficiency, and cervical screening are offered during the first visit. The ANC visit at 16–19 weeks of gestational age offers a fetal anatomy scan as well as a review of the tests recommended at the first ANC visit to plan more personalized care. An ANC visit at 20–27 weeks of gestational age tests for hyperglycemia as well as proteinuria for women who have clinical indications of pre-eclampsia. Tests for anemia, blood group, and antibodies are conducted during the ANC visit at 28 weeks of gestation. An ANC visit at 29–34 weeks of gestational age offers a 32-week ultrasound for women whose placenta extended over the internal cervical os during the 18-to-20-week scan. ANC visits after 35 weeks of gestation start examining fetal malpresentation.

While ANC is a well-established intervention for pregnancy care, existing research on ANC and perinatal outcomes has primarily focused either on the number of visits or on the timing of the first ANC visit. In addition, how the number of ANC visits impacts differently for women with early versus late entry into ANC has been less researched, and understanding this relationship has important public health policy implications. This study, therefore, aimed to examine the impact of the number of visits and the timing of the first ANC visit on LBW and preterm birth by utilizing regional health data from New South Wales for the period of 2011–2020. The findings from this study may be beneficial for the development and expansion of strategies for evidence-based maternity services to improve birth outcomes.

## 2. Materials and Methods

### 2.1. Data Source and Sample Composition

The source of the data utilized in this study was perinatal data that were collected as part of a population-based surveillance system for all births that take place across the state of New South Wales. The data collection has continuously operated since 1990, and the data are collected by attending midwives or medical practitioners. The aim of this data collection is to inform policy and programs for improved maternal and newborn health outcomes. The perinatal data include information on maternal sociodemographics, maternal health conditions, maternal health behaviors, perinatal care, and birth information, including discharge status, birth weight, and gestational age at birth. In SNSWLHD, 14,634 births took place during the period of 2011–2020. A total of 12,941 singleton live births were included in the analysis (Figure 1). For this study, mothers with residential addresses outside SNSWLHD, multiple births, stillborn babies, and babies without records of outcome and/or exposure variables were excluded.

### 2.2. Study Outcomes

The first outcome variable for this study was low birth weight. Low birth weight was dichotomously coded as ‘1′ if the measured birthweight was <2500 g or ‘0′ if the measured birthweight was ≥2500 g [14]. The second outcome variable for this study was preterm birth, coded as ‘1′ if the gestational age at birth was <37 weeks or ‘0′ if the gestational age at birth was ≥37 [15].

### 2.3. Exposure Variables

The first exposure variable for this study was based on the recommended number of ANC visit, as outlined in the Australian pregnancy care guideline [9]. The number of recommended ANC visits was categorized as ‘yes’ if women received at least ten visits for a first pregnancy or at least seven visits for a subsequent pregnancy or ‘no’ otherwise. The second exposure variable was the timing of the first ANC visit, categorized into three groups (first trimester, second trimester, and third trimester).

### 2.4. Confounders

The selection of confounding variables is based on previously published research on LBW and preterm birth [16,17,18,19] as well as information available in the perinatal data file. The confounding variables were grouped into community-level and socioeconomic factors, and maternal and newborn factors. Community-level factors included the hospital of birth (Cooma Health service in Cooma, Goulburn Base Hospital in Goulburn, Moruya District Hospital in Moruya, Queanbeyan Health Service in Queanbeyan, and South East Regional Hospital in Bega). All five hospitals included in this study provide level 3 maternity services from the antenatal period to birthing and the postnatal period for most women at 37–42 weeks of gestation. Collaborative maternity care is provided by health professionals such as midwives, GP obstetricians, and/or specialist obstetricians. The socioeconomic factor included in this study was an index for relative socioeconomic advantage and disadvantage (IRSAD) categorized into five quintiles (first, second, third, fourth, and fifth). The Australian Bureau of Statistics produces a socioeconomic index for areas (SEIFA) that scores areas according to relative socioeconomic advantage and disadvantage [20]. IRSAD scores are broadly based on people’s access to material and social resources and their ability to participate in society, such as occupied private dwellings, education, and employment. An area with a high score (the top 20%) indicates relatively high levels of socioeconomic advantage, whereas an area with a low score (the bottom 20%) indicates relatively high levels of socioeconomic disadvantage. Furthermore, an area with a high score represents many households with high incomes or more people with skilled occupations and less households with lower incomes or few people in unskilled occupations. Similarly, an area with a low score represents many households with low incomes or many people with unskilled occupations and few households with high incomes or few people in skilled occupations. The maternal factors were maternal age (20–34 years, <20 years, or 35+ years), Aboriginal status (non-Aboriginal or Aboriginal), maternal hypertension (no or yes), maternal diabetes (no or yes), and tobacco smoking (no or yes), whereas the newborn factors were newborn Aboriginal status (non-Aboriginal or Aboriginal), the five-minute Apgar score (<7 Apgar or ≥7 Apgar), newborn sex (male or female), and mode of birth (vaginal or Caesarean).

### 2.5. Statistical Analysis

Data analysis was conducted using STATA version 17.0 (StataCorp, College Station, TX, USA). The prevalences of LBW and preterm birth, according to the study factors, were estimated. A logistic regression was used to examine the associations between each study factor and the primary outcomes. Multivariable analyses were performed to examine the independent impacts of the recommended number of visits and the timing of the first ANC visit that controlled potential confounders.

A staged hierarchical technique [21] was used for multivariable analyses. As part of a staged technique, community-level and socioeconomic factors (including the year of birth) were analyzed first with manual backward elimination to remove statistically nonsignificant factors. Maternal and newborn factors were then added with significant community-level and socioeconomic factors with backward elimination to remove nonsignificant factors. In the third stage, exposure variables were examined separately with variables that were significantly associated with community-level and socioeconomic factors, and maternal and newborn factors. The significance level was set at 0.05, and variables that were significant in the third stage are reported in the manuscript. In the final model, collinearity was tested. Separate stratified analyses were performed to examine the effect sizes of the recommended number of ANC visits on LBW and preterm birth for women who made their first ANC visit in different stages of pregnancy (first trimester, second trimester, and third trimester).

### 2.6. Ethical Consideration

This study was approved by the Greater Western Human Research Ethics Committee (approval number: 2021/ETH10989, date of approval: 31 August 2021, protocol number: SNSWLHD 3_3_2021) and the Aboriginal Health and Medical Research Council Ethics Committee (approval number: 1998/22, date of approval: 26 September 2022, protocol number: SNSWLHD 3_3_2021). This study was authorized by the Southern New South Wales Local Health District (authorization number: 2022/STE01605).

## 3. Results

### 3.1. Characteristics of Study Sample

The largest group (27.6%) in the study sample was from the Queanbeyan Health Service (Table 1). The sample of women who were most disadvantaged (fifth quintile) was almost double (23.8%) compared to the women who were most advantaged (first quintile). More than 6% of the mothers included in this study were from an Aboriginal background. Smoking during pregnancy was reported to be 17.5%. Less than half of the study population made their first ANC visit during the first trimester, and almost 16% of the sample did not have the recommended number of ANC visits.

Over the study period, 338 (2.6%) LBW and 368 (2.8%) preterm births occurred in SNSWLHD. The prevalences of LBW and preterm birth among young mothers (<20 years of age) were higher (5.4% and 5.6%, respectively). LBW and preterm birth among babies with Apgar scores < 7 were substantially higher than those with scores ≥ 7. The percentage of LBW among mothers who reported diabetes was slightly lower (2.0%). In contrast, preterm birth among diabetic mothers was reported to be higher (3.6%).

### 3.2. Impact of Number and Timing of ANC on LBW

Multivariable analyses revealed that mothers who received the recommended number of ANC visits were 45% less likely to have LBW (aOR = 0.55 (95% CI: 0.42, 0.73)) compared to those who did not receive the recommended number of ANC visits (Table 2). The evidence also suggested that the timing of the first ANC visit was not significantly associated with LBW. However, when the number of recommended ANC visits was stratified by the timing, the results indicated that having the recommended number with early entry to ANC (first trimester) gave the greatest protection against LBW (aOR = 0.44 (95% CI: 0.26, 0.63)) (Figure 2).

### 3.3. Impact of Number and Timing of ANC on Preterm Birth

As shown in Table 3, mothers who received the recommended number of ANC visits had a 62% lower likelihood of preterm birth (aOR = 0.38 (95% CI: 0.30, 0.49)) compared to those who did not receive the recommended number of ANC visits. No significant association was observed between preterm birth and the timing of ANC. When the number of ANC visits was stratified by the timing, the results indicated that having the recommended number with early entry to ANC (first trimester) gave the greatest protection against preterm birth (aOR = 0.33 (95% CI: 0.21, 0.45)) (Figure 3).

## 4. Discussion

This study identified that the risk of LBW and preterm birth was reduced significantly in babies whose mothers reported an adequate number of visits along with early entry (first trimester) into ANC during 2011–2020. We also found that LBW and preterm birth differed significantly based on medical, social marginalization, and modifiable lifestyle behaviors. The most socioeconomically disadvantaged women had a higher likelihood of LBW and preterm birth. We observed a significant increase in the risk of LBW among Aboriginal mothers and those who reported tobacco smoking. This study revealed that diabetes affects the length of gestational age. Diabetic mothers were more prone to preterm birth compared to their non-diabetic counterparts.

The main strength of this study was our sample, which allowed us to examine the effect size of the recommended number of visits compared to the different timings of the first ANC visit while controlling for a range of confounding factors. The sample was selected (as described in Figure 1) from a cohort of all births that took place in SNSWLHD, which maximized the internal validity of this research. Even though some observations were excluded from this study, they were very low in number, and the probability of non-response bias is negligible. The measurement of outcomes and exposure were performed by trained clinicians with a greater level of uniformity over time and across the local health district, which minimized the measurement bias. Because of similarities in population diversity and geography, findings from this study may be extrapolated to other rural and regional LHDs of New South Wales. However, this study is also limited in several ways, and findings need to be interpreted with caution. Unmeasured confounders that were previously found to be associated with LBW or preterm birth included the maternal body mass index [22], subclinical hypothyroidism in pregnancy [23], and the prolonged prelabor rupture of membranes [17]. The small sample size in the stratified analysis for the third trimester may account for a large effect size with a wider confidence interval. Information on tobacco smoking was self-reported, which may account for potential information bias.

Early ANC is a foundation for optimal perinatal care. It helps clinicians to ascertain an accurate gestational age; identify pre-existing health conditions, obstetric complications, and fetal anomalies; and assess lifestyle and behavioral issues that adversely affect perinatal health [9]. Early ANC is also beneficial for multidisciplinary care providers (such as midwives, obstetricians, and Aboriginal and Torres Strait Islander health professionals) to quickly intervene with medical and health promotion activities [24]. Our findings emphasize that, while early entry into ANC is vital for optimal benefits, early entry should be complemented by a continuity of care via an adequate number of visits. In the third stage of our regression modeling, a significant impact of an adequate number of ANC visits on LBW and preterm birth was observed, consistent with previous studies [25,26,27]. However, contrary to research conducted in China [26] and Belgium [27], the timing of the first ANC visit in our study lacked significance. The statistically non-significant association between the timing of the first ANC visit and adverse perinatal outcomes may be because 46.1% of women who did not have an adequate number of ANC visits were those who had their first ANC visit during the first trimester. When the analysis of the ANC number was stratified by the timing of the first ANC visit, a greater level of protection was observed among mothers with early entry along with an adequate number of visits.

Socioeconomic inequality in LBW and preterm birth was evident in this study. An elevated risk of LBW or preterm birth among mothers with Aboriginal backgrounds and those with relative socioeconomic disadvantage may be due to a lower uptake of ANC linked to cultural and financial barriers, including transport and the opportunity cost of missing work [28,29]. Brown et al. highlighted the importance of culturally competent services while delivering midwifery care for Aboriginal women [30], and the early uptake of ANC (first trimester) among Aboriginal women was evident in Aboriginal-controlled community birthing programs [31]. The expansion of tailored community-based outreach ANC closer to those who need them most in regional NSW could be an important strategy to increase the coverage of ANC and subsequently prevent adverse perinatal outcomes. Race-based discrimination towards the Aboriginal population has been discussed as a barrier to health care access and utilization [32], and the National Aboriginal and Torres Strait Islander Health Workforce Strategic Framework [33] highlights that more investment is required in the indigenous health workforce to provide culturally safe and responsive health care for the Aboriginal population.

Smoking during pregnancy is a well-known avoidable risk behavior for adverse perinatal outcomes. Similar to the finding from a recent meta-analysis of observational studies [34], this study revealed that mothers who reported smoking during pregnancy were more likely to have LBW. Research has clearly documented the effects of harmful chemicals on fetal growth and development [35,36,37]. Similarly, mothers with diabetes were at higher risk of preterm birth, and this finding was consistent with a recent Western Australian study [38].

## 5. Conclusions

This study concludes that having early entry into ANC (first trimester), along with the recommended number of ANC visits, gave greater protection against LBW and preterm birth. In addition, there was no significant difference in LBW or preterm birth among mothers who attended recommended number of ANC visits but had late entry into ANC (third trimester). Our findings suggest that, to improve perinatal health, efforts should be made to promote both the early entry as well as the recommended number of visits. Such a focus is most likely to benefit women who report smoking during pregnancy, socioeconomically disadvantaged women, and those with Aboriginal backgrounds.

## Figures and Tables

**Figure 1 ijerph-20-00977-f001:**
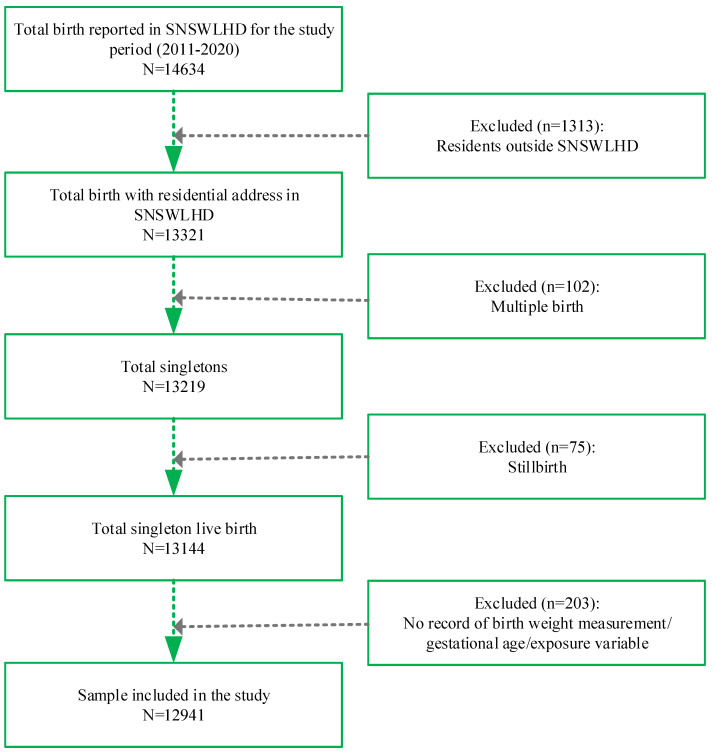
Selection of the sample for the period of 2011–2020 in the Southern New South Wales Local Health District (SNSWLHD).

**Figure 2 ijerph-20-00977-f002:**
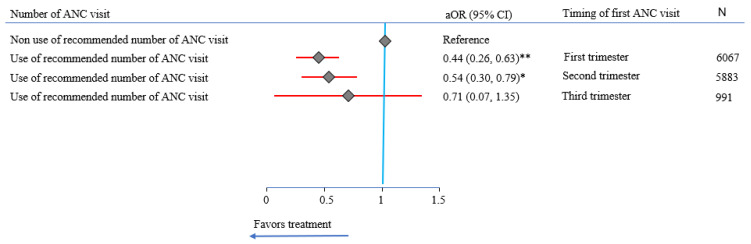
Impact of recommended number of antenatal care (ANC) visits on low birth weight (LBW) by the timing of the first visit in the Southern New South Wales Local Health District, determined by multivariate analysis, 2011–2020. **: *p*-value < 0.001; *: *p*-value <0.05; CI: confidence interval; aOR: adjusted odds ratio, adjusted for year of birth, hospital of birth, wealth quintile, maternal age, maternal Aboriginal status, maternal hypertension, maternal diabetes, tobacco smoking, newborn Aboriginal status, Apgar score, newborn sex, mode of birth, and preterm birth; number of cases included in multivariable analysis: 12,528.

**Figure 3 ijerph-20-00977-f003:**
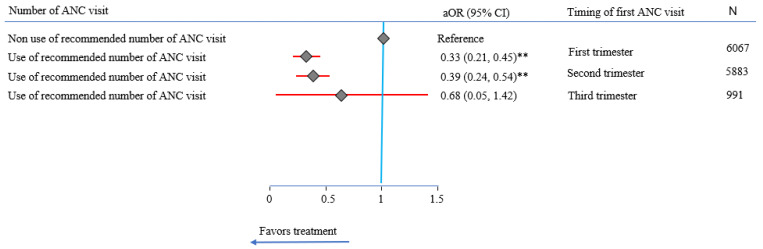
Impact of recommended number of antenatal care (ANC) visits on preterm birth by the timing of the first visit in the Southern New South Wales Local Health District, determined by multivariate analysis, 2011–2020. **: *p*-value < 0.001; CI: confidence interval; aOR: adjusted odds ratio, adjusted for year of birth, hospital of birth, wealth quintile, maternal age, maternal Aboriginal status, maternal hypertension, maternal diabetes, tobacco smoking, newborn Aboriginal status, Apgar score, newborn sex, mode of birth, and birth weight; number of cases included in multivariable analysis: 125,284.

**Table 1 ijerph-20-00977-t001:** Characteristics of the study sample and the number and prevalence of low birth weight and preterm birth by study factors in the Southern New South Wales Local Health District (2011–2020).

Study Factors	Study Sample	Low Birthweight		Preterm Birth	
Community-level and socioeconomic factors	N (% ^ϕ^)	n	% (95% CI)	n	% (95% CI)
Hospital of birth					
Cooma Health Service	1469 (11.4)	36	2.5 (1.8, 3.4)	34	2.3 (1.7, 3.2)
Goulburn Base Hospital	2819 (21.8)	95	3.4 (2.8, 4.1)	113	4.0 (3.3, 4.8)
Moruya District Hospital	2750 (21.3)	87	3.2 (2.6, 3.9)	91	3.3 (2.7, 4.0)
Queanbeyan Health Service	3569 (27.6)	61	1.7 (1.3, 2.2)	67	1.9 (1.5, 2.4)
South East Regional Hospital	2334 (18.0)	59	2.5 (2.0, 3.2)	63	2.7 (2.1, 3.4)
IRSAD quintile (N = 12,927)					
First (most advantaged)	1571 (12.1)	18	1.1 (0.7, 1.8)	24	1.5 (1.0, 2.3)
Second	854 (6.6)	16	1.9 (1.2, 3.0)	21	2.5 (1.6, 3.7)
Third	2462 (19.0)	47	1.9 (1.2, 3.0)	49	2.0 (1.5, 2.6)
Fourth	4956 (38.3)	138	2.8 (2.4, 3.3)	142	2.9 (2.4, 3.4)
Fifth (most disadvantaged)	3084 (23.8)	119	3.9 (3.2, 4.6)	132	4.3 (3.6, 5.1)
Maternal and newborn factors					
Maternal age					
20–34 years	10,243 (79.2)	255	2.5 (2.2, 2.8)	275	2.7 (2.4, 3.0)
<20 years	570 (4.4)	31	5.4 (3.9, 7.6)	32	5.6 (4.0, 7.8)
35+ years	2128 (16.4)	52	2.4 (1.9, 3.2)	61	2.9 (2.2, 3.7)
Maternal Aboriginal status (N = 12,907)					
Non-Aboriginal	12,077 (93.3)	286	2.4 (2.1, 2.7)	327	2.7 (2.4, 3.0)
Aboriginal	830 (6.4)	52	6.3 (4.8, 8.1)	41	4.9 (3.7, 6.6)
Maternal hypertension					
No	12,451 (96.2)	325	2.6 (2.3, 2.9)	362	2.9 (2.6, 3.2)
Yes	490 (3.8)	13	2.7 (1.5, 4.5)	6	1.2 (0.6, 2.7)
Maternal diabetes					
No	12,128 (93.7)	322	2.7 (2.4, 3.0)	339	2.8 (2.5, 3.1)
Yes	813 (6.3)	16	2.0 (1.2, 3.2)	29	3.6 (2.5, 5.1)
Tobacco smoking (N = 12,909)					
No	10,644 (82.3)	170	1.6 (1.4, 1.9)	248	2.3 (2.1, 2.6)
Yes	2265 (17.5)	163	7.2 (6.2, 8.3)	115	5.1 (4.2, 6.1)
Newborn Aboriginal status (N = 12,692)					
Non-Aboriginal	11,572 (89.4)	264	2.3 (2.0, 2.6)	311	2.7 (2.4, 3.0)
Aboriginal	1120 (8.7)	69	6.2 (4.9, 7.7)	51	4.6 (3.5, 5.9)
Apgar score (N = 12,847)					
≥7 Apgar	12,663 (97.9)	303	2.4 (2.1, 2.7)	332	2.6 (2.4, 2.9)
<7 Apgar	184 (1.4)	26	14.1 (9.8, 19.9)	28	15.2 (10.7, 21.2)
Newborn sex (N = 12,940)					
Male	6661 (51.5)	144	2.2 (1.8, 2.5)	193	2.9 (2.5, 3.3)
Female	6279 (48.5)	194	3.1 (2.7, 3.5)	175	2.8 (2.4, 3.2)
Mode of birth					
Vaginal	9488 (73.3)	241	2.5 (2.2, 2.9)	258	2.7 (2.4, 3.1)
Caesarean	3453 (26.7)	97	2.8 (2.3, 3.4)	110	3.2 (2.6, 3.8)
Exposure variables					
Timing of first ANC visit					
First trimester	6067 (46.9)	145	2.4 (2.0, 2.8)	165	2.7 (2.3, 3.2)
Second trimester	5883 (45.5)	160	2.7 (2.3, 3.2)	174	3.0 (2.6, 3.4)
Third trimester	991 (7.7)	33	3.3 (2.4, 4.6)	29	2.9 (2.0, 4.2)
Recommended number of ANC visits					
Yes	10,902 (84.2)	208	1.9 (1.7, 2.2)	221	2.0 (1.8, 2.3)
No	2039 (15.8)	130	6.4 (5.4, 7.5)	147	7.2 (6.2, 8.4)
Total	12,941 (100)	338	2.6 (2.4, 2.9)	368	2.8 (2.6, 3.1)

^ϕ^ Column percentage and percentage does not add up to 100 because of missing values. IRSAD: index for relative socioeconomic advantage and disadvantage; ANC: antenatal care; CI: confidence interval; maternal diabetes includes 794 cases of gestational diabetes and 19 cases of pre-existing diabetes.

**Table 2 ijerph-20-00977-t002:** Unadjusted and adjusted odds ratios (aORs) and 95% confidence intervals (CIs) for study variables and low birth weight in the Southern New South Wales Local Health District, 2011–2020.

Confounding Variables	uOR (95% CI)	*p*-Value	aOR (95% CI)	*p*-Value
Year of birth (2011–2020)	0.97 (0.93, 1.00)	0.083		
Hospital of birth				
Cooma Health Service	Reference			
Goulburn Base Hospital	1.39 (0.94, 2.05)	0.100		
Moruya District Hospital	1.30 (0.88, 1.93)	0.190		
Queanbeyan Health Service	0.69 (0.46, 1.05)	0.080		
South East Regional Hospital	1.03 (0.68, 1.57)	0.880		
IRSAD quintile (N = 12,927)				
First (most advantaged)	Reference		Reference	
Second	1.65 (0.84, 3.25)	0.150	1.60 (0.76, 3.38)	0.216
Third	1.68 (0.97, 2.90)	0.060	1.42 (0.77, 2.64)	0.261
Fourth	2.47 (1.51, 4.05)	<0.001	1.73 (0.99, 3.01)	0.053
Fifth (most disadvantaged)	3.46 (2.10, 5.71)	<0.001	1.88 (1.07, 3.31)	0.029
Maternal age				
20–34 years	Reference			
<20 years	2.25 (1.54, 3.30)	<0.001		
35+ years	0.98 (0.73, 1.33)	0.900		
Maternal Aboriginal status (N = 12,907)				
Non-Aboriginal	Reference		Reference	
Aboriginal	2.76 (2.03, 3.74)	<0.001	1.60 (1.08, 2.37)	0.019
Maternal hypertension				
No	Reference			
Yes	1.02 (0.58, 1.78)	0.950		
Maternal diabetes				
No	Reference			
Yes	0.74 (0.44, 1.22)	0.240		
Tobacco smoking (N = 12,909)				
No	Reference		Reference	
Yes	4.78 (3.83, 5.95)	<0.001	3.67 (2.82, 4.77)	<0.001
Newborn Aboriginal status (N = 12,692)				
Non-Aboriginal	Reference			
Aboriginal	2.81 (2.14, 3.69)	<0.001		
Apgar score (N = 12,847)				
≥7 Apgar	Reference		Reference	
<7 Apgar	6.71 (4.37, 10.32)	<0.001	3.25 (1.86, 5.69)	<0.001
Newborn sex (N = 12,940)				
Male	Reference		Reference	
Female	1.44 (1.16, 1.80)	<0.001	1.75 (1.36, 2.25)	<0.001
Mode of birth				
Vaginal	Reference			
Caesarean	1.11 (0.87, 1.41)	0.400		
Preterm birth				
No	Reference		Reference	
Yes	39.68 (30.81, 51.10)		28.7 (21.4, 38.47)	<0.001
Exposure variables				
Timing of first ANC visit				
First trimester	Reference			
Second trimester	1.14 (0.91, 1.43)	0.250		
Third trimester	1.41 (0.96, 2.07)	0.080		
Number of recommended ANC visits				
No	Reference		Reference	
Yes	0.29 (0.23, 0.36)	<0.001	0.55 (0.42, 0.73)	<0.001

IRSAD: index for relative socioeconomic advantage and disadvantage; ANC: antenatal care; number of cases included in multivariable analysis: 12,528.

**Table 3 ijerph-20-00977-t003:** Unadjusted and adjusted odds ratios (aORs) and 95% confidence intervals (CIs) for study variables and preterm birth in the Southern New South Wales Local Health District, 2011–2020.

Confounding Variables	uOR (95% CI)	*p*-Value	aOR (95% CI)	*p*-Value
Year of birth (2011–2020)	0.95 (0.92, 0.99)	0.01	0.95 (0.92, 0.99)	0.021
Hospital of birth				
Cooma Health Service	Reference			
Goulburn Base Hospital	1.76 (1.19, 2.6)	<0.001		
Moruya District Hospital	1.44 (0.97, 2.15)	0.070		
Queanbeyan Health Service	0.81 (0.53, 1.23)	0.320		
South East Regional Hospital	1.17 (0.77, 1.79)	0.460		
IRSAD quintile (N = 12,927)				
First (most advantaged)	Reference		Reference	
Second	1.63 (0.9, 2.94)	0.110	1.19 (0.62, 2.28)	0.601
Third	1.31 (0.8, 2.14)	0.280	1.03 (0.61, 1.76)	0.905
Fourth	1.9 (1.23, 2.94)	<0.001	1.39 (0.87, 2.21)	0.163
Fifth (most disadvantaged)	2.88 (1.86, 4.47)	<0.001	1.83 (1.14, 2.93)	0.012
Maternal age				
20–34 years	Reference			
<20 years	2.16 (1.48, 3.14)	<0.001		
35+ years	1.07 (0.81, 1.42)	0.640		
Maternal Aboriginal status (N = 12,907)				
Non-Aboriginal	Reference			
Aboriginal	1.87 (1.34, 2.6)	<0.001		
Maternal hypertension				
No	Reference			
Yes	0.41 (0.18, 0.93)	0.030		
Maternal diabetes				
No	Reference		Reference	
Yes	1.29 (0.87, 1.89)	0.200	1.68 (1.08, 2.59)	0.02
Tobacco smoking (N = 12,909)				
No	Reference			
Yes	2.24 (1.79, 2.81)	<0.001		
Newborn Aboriginal status (N = 12,692)				
Non-Aboriginal	Reference			
Aboriginal	1.73 (1.28, 2.34)	<0.001		
Apgar score (N = 12,847)				
≥7 Apgar	Reference		Reference	
<7 Apgar	6.67 (4.39, 10.11)	<0.001	3.38 (1.99, 5.74)	<0.001
Newborn sex (N = 12,940)				
Male	Reference			
Female	0.96 (0.78, 1.18)	0.710		
Mode of birth				
Vaginal	Reference			
Caesarean	1.18 (0.94, 1.48)	0.160		
Low birth weight				
No	Reference		Reference	
Yes	39.68 (30.81, 51.1)	<0.001	28.04 (21.28, 36.94)	<0.001
Exposure variables				
Timing of first ANC visit				
First trimester	Reference			
Second trimester	1.09 (0.88, 1.35)	0.430		
Third trimester	1.08 (0.72, 1.61)	0.710		
Number of recommended ANC visits				
No	Reference		Reference	
Yes	0.27 (0.21, 0.33)	<0.001	0.38 (0.30, 0.49)	<0.001

IRSAD: index for relative socioeconomic advantage and disadvantage; ANC: antenatal care; number of cases included in multivariable analysis: 12,528.

## Data Availability

Access to the data used in this study was in accordance with the research protocol submitted to the Human Research Ethics Committees. For data inquiries, please contact the Greater Western Human Research Ethics Committee (GWHREC) (postal address: P.O. Box 143, 39 Hampden Park Road, Bathurst, Australia 2795; telephone: +61-02-6330-5948).

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
