# Peer review of "Impact of Antenatal Care on Perinatal Outcomes in New South Wales, Australia: A Decade-Long Regional Perspective"

_ijerph, 2023, doi:10.3390/ijerph20020977_

Round 1
Reviewer 1 Report
Dear Authors,
Yours results did not suprised me.
My comments:
1. What does mean "reference" in tables and why was used so often?
2. "Wealth Quintile" and "IRSAD quintile" mean the same?
3. Too much tables not well described...
4. How is defined preterm birth in Australia?
5. If you started using abbreviations as LBW, you should use it in all text like you used ANC.
6. Why 10 visits in first pregnancy and 7 during next pregnancies?
7. You have high percentage of smoking women... why?
8. If you use abbreviations in tables or figures, they shoud be described in legend uder figure or table
Author Response
28/12/2022
International Journal of Environmental Research and Public Health
MDPI AG, St. Alban-Anlage 66
4052 Basel, Switzerland
Subject: Response to reviewer comments: Impact of antenatal care on perinatal outcomes in New South Wales, Australia: a decade long regional perspective (Manuscript ID: ijerph-2083735)
Special Issue " Prenatal Wellbeing and Maternal and Child Health Outcomes"
Dear Reviewer,
On behalf of all co-authors, I would like to thank you for your valuable comments, and suggestions that substantially helped to improve the quality of our manuscript entitled “Impact of antenatal care on perinatal outcomes in New South Wales, Australia: a decade long regional perspective (Manuscript ID: ijerph-2083735).
We are submitting the revised version of the manuscript for your consideration. Response to your comments, and details of the changes are provided below.
Comment: What does mean "reference" in tables and why was used so often?
Response: We used reference category so that the odds ratios for multi-category predictor variable could be compared to the reference group. This is in line with the journal’s guideline.
Comment: "Wealth Quintile" and "IRSAD quintile" mean the same?
Response: Thank you for indicating this. Wealth quintile or Index for Relative Socioeconomic Advantage and Disadvantage (IRSAD) are used to measure household socioeconomic status. We have revised the manuscript and used ‘IRSAD quintile’ throughout the manuscript for consistency.
Comment: Too much tables not well described...
Response: We have included three key tables. Table 1 provides the characteristics of study population; whereas table 2, and table 3 are multivariate regression models for each of the outcome (low birth weight, and preterm birth respectively). We believe we have sufficiently described key findings of each table in the form of sub-headings under ‘Results’ section (table 1 is described under sub-heading: 3.1. Characteristics of study sample, table 2 is described under sub-heading: 3.2. Impact of number and timing of ANC on LBW, and table 3 is described under sub-heading: 3.3. Impact of number and timing of ANC on preterm birth). The description of tables in this manuscript is consistent with our past literature [1-3].
Comment: How is defined preterm birth in Australia?
Response: Australia follows the World Health Organization (WHO) definition of preterm birth (https://www.health.gov.au/resources/pregnancy-care-guidelines/part-d-clinical-assessments/risk-of-preterm-birth) defined as birth before 37 completed weeks of pregnancy. We have used the WHO definition of preterm birth for this study.
Comment: If you started using abbreviations as LBW, you should use it in all text like you used ANC.
Response: Thank you for highlighting this. We have revised the manuscript as per reviewer’s comment.
Comment: Why 10 visits in first pregnancy and 7 during next pregnancies?
Response: The number of antenatal care visits in this study is based on Australian Pregnancy Care Guideline, approved by National Health and Medical research Council. The first pregnancy has always more to learn and adopt for first time pregnant women. This includes but not limited to health education such as breastfeeding, tobacco smoking, and understanding referral processes. A lot of these processes may not necessarily require the same amount of time/level of intensity for subsequent pregnancy. Therefore, the number of ANC visits vary between first and subsequent pregnancy.
Comment: You have high percentage of smoking women... why?
Response: The percentage of smoking reported in this study represents study time frame (2011-2021). However, smoking rate over the last decade in Southern New South Wales Local Health District has decreased. For reviewer’s purpose we have included Figure 1.
Figure 1. Percentage of smoking in pregnancy in Southern New South Wales Local Health District; Source: NSW Perinatal Data Collection (SAPHaRI). Centre for Epidemiology and Evidence, NSW Ministry of Health.
Comment: If you use abbreviations in tables or figures, they shoud be described in legend uder figure or table
Response: Thank you for the comment. We have now described all abbreviations in tables or figures into legends/footnotes (Please see revised manuscript).
References
- Dhami, M.V., et al., Regional analysis of associations between infant and young child feeding practices and Diarrhoea in Indian children. International Journal of Environmental Research and Public Health, 2020. 17(13): p. 4740.
- Ghimire, P.R., et al., Smoking Cessation during the Second Half of Pregnancy Prevents Low Birth Weight among Australian Born Babies in Regional New South Wales. International Journal of Environmental Research and Public Health, 2021. 18(7): p. 3417.
- Ghimire, P.R., et al., Under-five mortality and associated factors: evidence from the Nepal Demographic and Health Survey (2001–2016). International journal of environmental research and public health, 2019. 16(7): p. 1241.
Yours sincerely,
Dr. Pramesh Raj Ghimire
Priority Populations | Southern NSW Local Health District
Level 3, 34 Lowe St QUEANBEYAN, New South Wales 2620
Tel: +61261507560 |Mob: +61436852496
Email: Pramesh.Ghimire@health.nsw.gov.au
Reviewer 2 Report
Thank you for opportunity to review the manuscript "Impact of antenatal care on perinatal outcomes in New South Wales, Australia: a decade long regional perspective". As a specialist in perinatal epidemiology, I agree compliance of women to ANC is an important part in prevention of adverse perinatal outcomes.
In spite of importance the topic the manuscript has some methodological issues that need to be clarifying:
Introduction
1. Introduction section should give the readers opportunity to get more information on system of ANC in Australia; the authors should not just use the references, but describe the most important points from national guidelines.
2. Method section should contain more information on the source of data, process of data collection and proportion of missing data. There is also some inconsistency in numbers. At fig 1: Sample included in the study, N=12941, but latter in text and tables we can see different numbers from 12940 to 12847. Does it mean that number of outcomes included were different for different variable? If so, it is not common practice and should be addressed in methods. It also should be mentioned a number of cases included in multivariable analysis.
3. Authors use IRSAD score as a measure of socioeconomic wellbeing. It is not a common practice, thus it should be described in details.
4. Why did you not take in to account other SES factors, like education, civil status of women? According to many studies they considered as an independent risk factors of LBW and PB.
5. All categorical variables used in model should be presented in Methods with categorization used with explanation of reference category choice.
6. In tables 1-3 you use Hospital of birth variable, but most of the potential readers will be interesting in level of the hospital, nor the names. I consider you should rearrange this variable as a level of hospital
7. How maternal comorbidities were defined? As for maternal diabetes, do you include all forms or gestational diabetes only?
8. What was a source of data on maternal smoking, is any possibility for informational bias here?
Results
9. I suggest to add statistical comparison of PB and LBW proportion in table 1
10. Some of variables, which had significant association with PB and LBW at univariate analysis were not selected for the final model during the staged hierarchical technique. I think possible explanation for this should be discussed.
11. Authors stated that internal validity of research is high, but did not discuss possibility for selection and informational biases. Generally, there is no information on limitation of the study in Discussion part.
Author Response
28/12/2022
International Journal of Environmental Research and Public Health
MDPI AG, St. Alban-Anlage 66
4052 Basel, Switzerland
Subject: Response to reviewer comments: Impact of antenatal care on perinatal outcomes in New South Wales, Australia: a decade long regional perspective (Manuscript ID: ijerph-2083735)
Special Issue " Prenatal Wellbeing and Maternal and Child Health Outcomes"
Dear Reviewer,
On behalf of all co-authors, I would like to thank you for your valuable comments, and suggestions that substantially helped to improve the quality of our manuscript entitled “Impact of antenatal care on perinatal outcomes in New South Wales, Australia: a decade long regional perspective (Manuscript ID: ijerph-2083735).
We are submitting the revised version of the manuscript for your consideration. Response to your comments, and details of the changes are provided below.
Introduction
Comment: Introduction section should give the readers opportunity to get more information on system of ANC in Australia; the authors should not just use the references, but describe the most important points from national guidelines.
Response: Thank you for the comment. We have revised the fourth paragraph of introduction section on ANC that highlights the most important points from national guidelines. The revised paragraph in line 65-89 now reads as:
“Quality Antenatal Care (ANC) is one of the core evidence-based interventions in the first 2000 days of human lifespan [13]. To provide quality antenatal ANC, the Australian pregnancy care guidelines recommend a schedule of ten visits for a first pregnancy without complications, whereas for subsequent uncomplicated pregnancies, a schedule of seven visits is recommended [9]. In addition, the recommended timing for the first ANC visit occurs within the first trimester of pregnancy. These recommendations were approved by National Health and Medical Research Council of Australia [9]. ANC is provided in both the community, and the hospital-based settings. Health professionals who contribute to antenatal care includes but not limited to midwives, general practitioners, obstetricians, and Aboriginal and Torres Strait Islander health practitioners. Apart from routine assessments such as height and weight, blood pressure, fetal growth, and fetal movements, each antenatal visit has distinct clinical importance to better manage pregnancy care. For example, the first ANC visit undertakes a comprehensive history including obstetric, medical and lifestyle behaviors such as smoking during pregnancy. Clinical assessments and testing such as human immunodeficiency virus (HIV), hepatitis B, hepatitis C, chromosomal anomalies, thyroid function, vitamin D deficiency, and cervical screening are offered in the first visit. The ANC visit during 16-19 weeks of gestational age will offer fetal anatomy scan, as well as review the tests recommended at first ANC visit to plan more personalized care. An ANC visit during 20-27 weeks of gestational age will test for hyperglycemia, as well as proteinuria for women who have clinical indications of pre-eclampsia. Tests for anemia, blood group and antibodies will be conducted in the ANC visit at 28 weeks gestation. An ANC visit during 29-34 weeks of gestational age will offer 32 weeks ultrasound for women whose placenta extended over the internal cervical os during the 18 to 20-week scan. ANC visits after 35 weeks gestation will start examining fetal malpresentation.”
Comment: Method section should contain more information on the source of data, process of data collection and proportion of missing data. There is also some inconsistency in numbers. At fig 1: Sample included in the study, N=12941, but latter in text and tables we can see different numbers from 12940 to 12847. Does it mean that number of outcomes included were different for different variable? If so, it is not common practice and should be addressed in methods. It also should be mentioned a number of cases included in multivariable analysis.
Response: Data source and sample composition section has been revised which now reads as:
“The source of data utilized in this study was perinatal data which are collected as part of a population-based surveillance system for all births that take place across the state of New South Wales. The data collection has been continuously operated since 1990, and the data are collected by attending midwives or medical practitioners. The aim of this data collection is to inform policy and programs for improved maternal and newborn health outcomes. Perinatal data collects information on maternal socio-demographic, maternal health condition, maternal health behaviors, perinatal care, and birth information including discharge status, birth weight and gestational age at birth. In SNSWLHD, 14634 birth took place during the period 2011-2020. A total of 12,941 singleton live birth was included in the analysis [Fig.1]. For this study, mothers with residential address outside SNSWLHD, multiple birth, stillborn babies, and babies without the record of outcome and/ exposure variables were excluded”
The sample included in this study is 12941. However, there are few missing values for variables such as IRSAD quintile, maternal Aboriginal status, tobacco smoking, newborn Aboriginal status, apgar score, and newborn sex. To highlight missing values for each of the variable, we have provided non-missing counts in the bracket, consistent with previously published studies [1, 2]. Based on reviewer’s comment, we have addressed this in the footnote of tables and figures from multivariate analysis (Please see footnotes of Table 2, Table 3, Figure 2, and Figure 3).
Comment: Authors use IRSAD score as a measure of socioeconomic wellbeing. It is not a common practice, thus it should be described in details.
Response: As per suggestion from reviewer, we have provided additional details for IRSAD variable, which in the revised manuscript (line 144-157) reads as:
“The socioeconomic factor included in this study is an Index for Relative Socio-economic Advantage and Disadvantage (IRSAD). Australian Bureau of Statistics produces Socio-Economic Index for Areas (SEIFA) that scores areas according to Relative Socio-Economic Advantage and Disadvantage [20]. IRSAD scores are broadly based on people’s access to material and social resources, and their ability to participate in society such as occupied private dwellings, education, and employment. An area with high score (the top 20%) indicates relatively high levels of socio-economic advantage, whereas an area with low score (the bottom 20%) indicates relatively high levels of socio-economic disadvantage. Furthermore, an area with high score represents many households with high incomes, or more people with skilled occupations; and less households with lower income, or few people in unskilled occupations. Similarly, an area with low score represents many households with low incomes, or many people with unskilled occupations; and few households with high incomes, or few people in skilled occupations.”
Comment: Why did you not take in to account other SES factors, like education, civil status of women? According to many studies they considered as an independent risk factors of LBW and PB.
Response: Very valid point indeed. This is a secondary data analyses, and SES factors such as education, or civil status of women are not separately available in the source data file utilized in this study. However, information on educational attainment is included in the IRSAD index, and we have highlighted this to describe IRSAD quintile in the revised method section (line 150).
Comment: All categorical variables used in model should be presented in Methods with categorization used with explanation of reference category choice.
Response: As per reviewer’s comment we have presented categorization of each categorical variables in our revised method section. The choice of reference category in this study was to have a logical comparison between the most normative groups, and for easier interpretation to suit local context. The way we have presented variables with categorization in our revised manuscript is consistent with previous literature [3].
Comment: In tables 1-3 you use Hospital of birth variable, but most of the potential readers will be interesting in level of the hospital, nor the names. I consider you should rearrange this variable as a level of hospital
Response: Thank you for the important comment. All five hospitals included in this study provide level 3 maternity services. We have highlighted this in the revised method section which reads as:
“Community level factor included hospital of birth (Cooma Health service in Cooma, Goulburn Base Hospital in Goulburn, Moruya District Hospital in Moruya, Queanbeyan Health Service in Queanbeyan, and South East Regional Hospital in Bega). All five hospitals included in this study provide level 3 maternity services from antenatal, birthing, to postnatal period for most women with 37-42 weeks of gestation. A collaborative maternity care is provided by health professionals such as midwives, GP obstetricians and/or specialist obstetricians.”
Comment: How maternal comorbidities were defined? As for maternal diabetes, do you include all forms or gestational diabetes only?
Response: We included all forms of diabetes given the number of pre-existing diabetes for the period (2011-2020) was very small (19 cases) as shown in the STATA output below. We have made this clear in the footnote of table 1.
Comment: What was a source of data on maternal smoking, is any possibility for informational bias here?
Response: Data on smoking during pregnancy are self-report. We have included potential informational bias in our discussion (please see line 273-274 of the revised manuscript).
Results
Comment: I suggest to add statistical comparison of PB and LBW proportion in table 1
Response: Thank you for the comment. For statistical comparison we have plotted confidence intervals around preterm birth and low birth weight prevalence in table 1.
Comment: Some of variables, which had significant association with PB and LBW at univariate analysis were not selected for the final model during the staged hierarchical technique. I think possible explanation for this should be discussed.
Response: Some of the variables which were significantly associated with PB and LBW at univariate analyses were eliminated during staged elimination process as highlighted in the manuscript (line 170-179), and this technique has been previously used [4-6]. We also tested for collinearity, and this has been mentioned in the revised manuscript under ‘Statistical analysis’ section (line 178-179 of the revised manuscript).
Comment: Authors stated that internal validity of research is high, but did not discuss possibility for selection and informational biases. Generally, there is no information on limitation of the study in Discussion part.
Response: This research included all birth that took place in Southern New South Wales Local Health District; and hence why, the selection bias was not considered for study limitation [3]. Based on reviewer’s comment we have included an additional study limitation on informational bias (please see line 273-274 of the revised manuscript).
References
- Ghimire, P.R., et al., Factors associated with perinatal mortality in Nepal: evidence from Nepal demographic and health survey 2001–2016. BMC pregnancy and childbirth, 2019. 19(1): p. 1-12.
- Ghimire, P.R., et al., Under-five mortality and associated factors: evidence from the Nepal Demographic and Health Survey (2001–2016). International journal of environmental research and public health, 2019. 16(7): p. 1241.
- Ghimire, P.R., et al., Smoking Cessation during the Second Half of Pregnancy Prevents Low Birth Weight among Australian Born Babies in Regional New South Wales. International Journal of Environmental Research and Public Health, 2021. 18(7): p. 3417.
- Dhami, M.V., et al., Regional analysis of associations between infant and young child feeding practices and Diarrhoea in Indian children. International Journal of Environmental Research and Public Health, 2020. 17(13): p. 4740.
- Akombi, B.J., et al., Stunting and severe stunting among children under-5 years in Nigeria: A multilevel analysis. BMC pediatrics, 2017. 17(1): p. 1-16.
- Ghimire, P.R., et al., Socio-economic predictors of stillbirths in Nepal (2001-2011). PloS one, 2017. 12(7): p. e0181332.
Yours sincerely,
Dr. Pramesh Raj Ghimire
Priority Populations | Southern NSW Local Health District
Level 3, 34 Lowe St QUEANBEYAN, New South Wales 2620
Tel: +61261507560 |Mob: +61436852496
Email: Pramesh.Ghimire@health.nsw.gov.au

Round 2
Reviewer 1 Report
Dear Authors,
I accept you reply and I am satisfied
Reviewer 2 Report
The authors clearly addressed all reviewer concerns and the manuscript has been revised and significantly improved.